# Urinary Medium-Chained Acyl-Carnitines Sign High Caloric Intake whereas Short-Chained Acyl-Carnitines Sign High -Protein Diet within a High-Fat, Hypercaloric Diet in a Randomized Crossover Design Dietary Trial

**DOI:** 10.3390/nu13041191

**Published:** 2021-04-03

**Authors:** Nadezda V. Khodorova, Annemarie Rietman, Douglas N. Rutledge, Jessica Schwarz, Julien Piedcoq, Serge Pilard, Els Siebelink, Frans J. Kok, Daniel Tomé, Marco Mensink, Dalila Azzout-Marniche

**Affiliations:** 1 AgroParisTech, Université Paris-Saclay, INRAE, UMR PNCA, 75005 Paris, France; nadezda.khodorova@agroparistech.fr (N.V.K.); julien.piedcoq@agroparistech.fr (J.P.); tome@agroparistech.fr (D.T.); 2Division of Human Nutrition & Health, Wageningen University, NL-6703 HD Wageningen, The Netherlands; annemarierietman@hotmail.com (A.R.); jessica.schwarz@outlook.com (J.S.); els.siebelink@wur.nl (E.S.); frans.kok@wur.nl (F.J.K.); marco.mensink@wur.nl (M.M.); 3AgroParisTech, Université Paris-Saclay, INRAE, UMR SayFood, 91300 Massy, France; douglas.rutledge@agroparistech.fr; 4Plateforme Analytique, Université de Picardie Jules Verne, 80039 Amiens, France; serge.pilard@u-picardie.fr

**Keywords:** dietary protein, postprandial response, lipid metabolism, glucose metabolism, short chain acyl-carnitine, medium chain acyl-carnitine, urinary metabolomics, plasma metabolomics

## Abstract

The western dietary pattern is known for its frequent meals rich in saturated fat and protein, resulting in a postprandial state for a large part of the day. Therefore, our aim was to investigate the postprandial glucose and lipid metabolism in response to high (HP) or normal (NP) protein, high-fat hypercaloric diet and to identify early biomarkers of protein intake and hepatic lipid accumulation. In a crossover design, 17 healthy subjects were randomly assigned to consume a HP or NP hypercaloric diet for two weeks. In parallel, a control group (CD; *n* = 10) consumed a weight-maintaining control diet. Biomarkers of postprandial lipid and glucose metabolism were measured in 24 h urine and in plasma before and following a meal challenge. The metabolic profile of urine but not plasma, showed increased excretion of ^13^C, carnitine and short chain acyl-carnitines after adaptation to the HP diet. Urinary excretion of decatrienoylcarnitine and octenoylcarnitine increased after adaptation to the NP diet. Our results suggest that the higher excretion of short-chain urinary acyl-carnitines could facilitate the elimination of excess fat of the HP diet and thereby reduce hepatic fat accumulation previously reported, whereas the higher excretion medium-chains acyl-carnitine could be early biomarkers of hepatic lipid accumulation.

## 1. Introduction

The western dietary pattern, rich in foods of animal origin with high saturated fat and protein content, has repeatedly been associated with an increased risk of developing metabolic syndrome and its associated dysfunctions including non-alcoholic fatty liver disease (NAFLD) [1,2]. A hypercaloric diet, especially rich in trans/saturated fat and cholesterol, as well as fructose-sweetened beverages, seem to increase visceral adiposity and stimulate hepatic lipid accumulation and progression into non-alcoholic steatohepatitis [1,3]. Therefore, adjusting nutrition habits may serve as a major route in preventing metabolic dysfunction. The development of biomarkers for early detection of lipid accumulation is hence an important task.

The influence of high dietary protein intake on metabolic dysfunction and weight loss remains controversial [4]. A high protein intake was associated with an increased risk of developing insulin resistance and the metabolic syndrome [5,6]. Ingestion of protein acutely reduces insulin action as measured with a hyperinsulinemic-euglycemic clamp [7]. However, other studies have shown that protein-enriched diets compared to standard protein diets can lead to improvements in biomarkers of metabolic syndrome (MetS) [8,9]. Along these lines, we and others have previously shown that increasing the protein content of the diet was effective in reducing intra-hepatic lipid content, fasting plasma triglycerides (TG) concentrations [9,10,11,12] and body fat mass [8,11,12,13,14]. These findings are particularly interesting in the context of a high caloric intake, the most important determinant of liver fat accumulation [1,15,16].

We have previously reported that, in healthy subjects, a high protein hypercaloric diet is associated with a lower fat accumulation in the liver compared to a normal protein hypercaloric diet [11]. As circulating metabolites during the postprandial period are particularly interesting when studying the effect of different diets on the metabolism, the objective of the present study was to characterize postprandial metabolism in these subjects in response to a different protein level in the diet and to identify early biomarkers of hepatic fat accumulation in plasma and/or urine. Therefore, we measured, after two weeks of adaptation to the different conditions, markers of postprandial lipid and glucose metabolism in response to a mixed-macronutrient meal challenge. In addition, urine and plasma metabolomics analysis were performed for: (i) the detection of biomarkers associated with the consumption of a hypercaloric high- or normal protein diet; (ii) the identification of early biomarkers of hepatic lipid accumulation.

## 2. Materials and Methods

### 2.1. Participants and Study Design

Twenty-nine healthy, young, lean, Caucasian, male and female volunteers participated in this previously described strictly-controlled dietary intervention carried out in the clinical trial facility of Wageningen University in the Netherlands [11]. Eligibility criteria, detailed experimental and control interventions, their administration, determination of sample size, randomization methods and blinding were published earlier [11]. The experimental protocol was approved by the Medical Ethics Committee of Wageningen University, The Netherlands. The trial was registered at https://clinicaltrials.gov/ct2/show/NCT01354626 as NCT01354626 (August 2011). The primary outcome measure of the trial, hepatic fat accumulation, as well as some of the secondary outcome measures as well as the way they were assessed were published earlier [11]. In the present report, the secondary outcome measures postprandial lipid metabolism and glucose homeostasis are discussed. In contrast to the analyses originally planned, stable isotope measurements were not carried out due to technical problems with the samples. Recruitment started in August 2011 and the study was completed in March 2012. The total dietary intervention lasted for 6 weeks (Figure 1).

All participants started with a two weeks run-in weight maintaining control diet (27.8 En% fat; 16.9 En% protein; 55.3 En% carbohydrates). Two participants dropped out of the study for personal reasons after the run-in period but before starting the intervention diets. Therefore, data of for a total of 27 participants are presented. After the run-in period, 17 randomly selected participants took part in the cross-over design intervention with two separate hypercaloric high-fat intervention periods (HD, overfed with 2MJ per day) of two weeks each. Volunteers in this group were randomly assigned to consume during each period either the normal-protein diet (NP; 39.4 En% fat; 15.4 En% protein; 45.2 En% carbohydrate) or the high-protein diet (HP; 37.7 En% fat; 25.7 En% protein; 36.6 En% carbohydrate). Participants started randomly with either the NP condition, or the HP condition and crossed to the other condition after two weeks. In parallel, a control group consumed the weight maintaining control diet for another four weeks (CD-group; *n* = 10). Measurements were performed at the end of each two-week period (Period 1 and Period 2 respectively, see Figure 1). More details on the dietary intervention can be found in [11].

### 2.2. Meal Challenge and Urine and Plasma Sampling

A meal challenge (MC) was performed at the end of each intervention period. After an overnight fast, all subjects received the same MC of 3 MJ with 40 En% fat (saturated fatty acids (SFA): 22.8 En%, monounsaturated fatty acids (MUFA): 9.9 En% and polyunsaturated fatty acids (PUFA): 1.2 En%), 15 En% protein and 45 En% carbohydrates, presented as a semi-liquid meal (ingredients: low fat yoghurt: 114.3 g; water: 121.3 g; low-fat curd cheese: 116.7 g; whipped cream: 102.7 g; strawberry syrup: 88.7 g; sugar: 24.5 g; whey protein powder: 15.2 g). The energy intake from the MC, related to body weight (BW), was 10.5 ± 1.29 kcal/kg BW (range from 8.7–13.6).

In the last 24 h before the meal challenge, the urine was collected and frozen at −80 °C until analysis. A fasted blood sample was drawn from an antecubital vein (t = −15, baseline), thereafter, a catheter was inserted into a vein on the dorsal side of the hand, in retrograde direction for sampling arterialized venous blood. To assure arterialization of the venous blood, the hand was put in a heated box and warmed to 55 °C for 15 min before each blood sampling. The catheter was kept patent by repeated flushing with 0.9% NaCl. At regular time points, from just before the MC (t = 0) and 30, 60, 120, 180, 240, 360 min after the MC, blood samples were collected in heparinized syringes. Plasma was separated and stored at −80 °C until analysis.

### 2.3. Biochemical Analyses

Plasma biochemical analysis was carried out on fasting and postprandial samples. Analyses of triglycerides (Randox, Crumlin, Dublin, Ireland), free fatty acid (FFA) (Randox, Crumlin, Dublin, Ireland), glycerol (Randox, Crumlin, Dublin, Ireland), β-hydroxybutyrate (β-HB) (Randox, Crumlin, Dublin, Ireland), cholesterol and HDL cholesterol (Beckman Coulter, Brea, CA, USA) were performed on an Olympus AU400 robot (Centre d’ Explorations Fonctionnelles—Imagerie (CEFI), Bichat, Paris, France). Glucose was analysed by the hexokinase method (Beckman Coulter, Brea, CA, USA) and insulin was measured by enzyme-linked immunosorbent assay (ELISA, Mercodia, Uppsala, Sweden).

### 2.4. ^13^C Enrichment in Urine

The ^13^C enrichment of urine samples was determined using isotope ratio mass spectrometer (Isoprime, GV Instruments, Manchester, UK) coupled with an elemental analyser (Vario Micro Cube, Elementar, Lyon, France). The ^13^C/^12^C ratios were measured in urine in reference to a calibrated CO_2_ gaz. The isotopic standard used was glutamic acid (USGS41). Enrichments were expressed as atom %.

### 2.5. Metabolomics Sample Preparation

One hundred micro-litres of each thawed 24 h urine sample were diluted with 900 μL of milliQ water/acetonitrile (90/10, *v*/*v*). For plasma deproteinization, 200 µL of fasting samples were mixed with 200 µL of cold methanol and stored at −20 °C overnight. The samples were then centrifuged and 100 µL of the supernatant was diluted with 900 μL of milliQ water/acetonitrile (90/10, *v*/*v*). All plasma and urine samples were then filtered using 0.2 nm Phenex-RC Syringe Filters with a cellulose membrane (Phenomenex, Le Pecq, France), transferred to a LC vial and either kept at −20 °C or stored at 4 °C if used within 1–2 days until analysis. Blank and quality control (QC) samples were injected every 10 analytical samples in order to control the possible instrument deviation. The blank samples contained acetonitrile and water (10/90, *v*/*v*) and QC samples contained the aliquots of all analyzed samples. All solvents used were liquid chromatography mass spectrometry (LC-MS) grade (Sigma-Aldrich; Saint Quentin Fallavier, France) and the aqueous solutions were prepared using purified Milli-Q water. The authentic standards of carnitine, acetyl-carnitine, propionyl-carnitine, butyryl-carnitine, 2-methylbutyroyl-carnitine, uric acid and 1-methylhistidine were purchased from Sigma-Aldrich.

### 2.6. Liquid Chromatography Mass Spectrometry (LC-MS) Analysis of Urine and Plasma

The LC-MS analysis was done using on an ultra-high performance liquid chromatograph (UPLC) Acquity H-Class system coupled to a Q-TOF Synapt G2 Si instrument (Waters Corporation, Milford, MA, USA).

The urine samples were analyzed using two chromatographic techniques: reverse-phase chromatography (RP) and hydrophilic interaction chromatography (HILIC) in positive electrospray ionization mode (ESI+). For the RP chromatography, an Acquity CSH C18 column (2.1 × 100 mm; 1.7 µm bead size; Waters) was used. Column temperature was set at 35 °C and the flow rate was 0/4 mL/min. The eluents A and B were 0.01% formic acid in water and 0.01% formic acid in acetonitrile, respectively. The gradient was as follows: an isocratic elution for 0.5 min of 5% B, then an increase of B at a linear rate to 95% at 10 min, with a re-equilibration for 1 min with 5% B and maintenance at 5% B until 16 min. For HILIC, an Acquity UPLC BEH Amide column (2.1 × 100 mm; 1.7 µm bead size; Waters) was used. Column temperature was 45°C and the eluents were in ACN/20 mM of pH 3.5 ammonium formate (50/50, *v*/*v*; A) and ACN/20 mM of pH 3.5 ammonium formate (90/10, *v*/*v*; B). The HILIC gradient was run at 0.6 mL/min and consisted of an isocratic elution for 2.5 min of 100% B, followed by a decrease of B at a linear rate to 0% in 10 min, then a re-equilibration for 1 min with 100% B and maintenance at 100% B until 15 min. Injection volume was 2 µL for both RP and HILIC.

The plasma samples were analyzed by RP chromatography using the same conditions as described for urines. The MS source conditions were as follows: capillary voltage at 3.0 kV; cone voltage at 20 V; source 20 V. The MS data were acquired from 50 to 1100 Da in centroid mode at 0.2 sec/scan. The leucine-enkephalin was used as a lock mass for mass accuracy correction. For MS/MS experiments performed on selected [M + H]+ ions, the parameters of collision induced dissociation (CID) parameters are indicated in Table 1.

### 2.7. LC-MS Data Processing and Analysis

Raw spectral data were processed using MarkerLynx (MassLynx V4.1, Waters) to obtain a matrix of detected features characterized by retention time (Rt) and *m*/*z*. The processing parameters were as follows: an extracted ion chromatogram (EIC) window of 0.03 Da, an intensity threshold of 500, a noise elimination of 8 and a retention time window of 0.2 min. The obtained matrix was transferred to Matlab version 7.6.0 (Mathworks Inc., Natick, MA, USA) for statistical evaluation of the data and chemometrics analysis by independent components–discriminant analysis (IC-DA).

The first step of data processing was to remove all insignificant features which appeared in blank samples with intensities greater than 1/10 the mean intensity of all features as well as those absent from QC samples. To reduce uncontrolled variations in global intensity due to instrumental factors, the signals were corrected using probabilistic quotient normalisation [17] based on the median of the acquired signals as the reference vector. With the aim of giving more weight to small but potentially significant changes in features, pareto scaling was applied. Pareto scaling [18] is a pretreatment method where the intensity of each feature is divided by the square root of its standard deviation. After this data pretreatment, the blanks were removed from the matrix and IC-DA was applied. The accurate mass and molecular formula prediction were screened for putative molecules in freely available databases (Metlin Metabolite, Human Metabolome Database, Chemspider, KEGG). The identification of the biomarker candidates was accomplished by comparison of MS/MS spectra with authentic standards (when available) and/or with those from earlier published data. The four levels described by the Metabolomics Standard Initiative were considered for metabolite identification [19].

Independent components analysis (ICA) is a method of blind source separation (BSS) [20] which assumes that each row of the data matrix is a weighted sum of source signals, the weights being proportional to the contribution of the corresponding source signals to that particular mixture. Neither the original source signals, nor their proportions in the observed signals are known. ICA extracts these sources underlying the observed signals, as well as their concentration in each mixture. The general model is given as: X = A.S, where X is the matrix of experimental data with *n* rows (the observed signals) and m columns (the data points in the signals), and A is the *n* × k matrix of proportions for k ICs. S is the k × m matrix of independent source signals (ICs). Independent components discriminant analysis (IC-DA) is an extension of ICA [20,21,22] that orients the extraction of the signals in the matrix, X, by concatenating it column-wise with a vector, y, characterizing the samples, and then performing ICA on the [X||y] matrix. In the case of IC-DA, the matrix of observed signals, X, is concatenated with a weighted binary-coded group membership matrix, corresponding to the information about the predefined groups of samples, as a means to orient the extraction of signals towards those that discriminate predefined groups of samples. The optimal number of ICs for IC-DA model is determined as the number that maximizes the F-value associated with the Wilks’ lambda characterizing the discrimination of the predefined groups based on the calculated proportions A. To determine whether the MS signals extracted by IC-DA are alone able to separate the samples into the predefined groups, new proportions, AX, for the ICs are then calculated using only the MS data (X) part of the extracted signals [23]. Permutation tests were then used to verify the validity of the observed discriminant models. ICA and IC-DA was performed using the Joint Approximation Diagonalization of Eigenmatrices (JADE) algorithm [24,25,26].

### 2.8. Statistical Analyses

All data are expressed as mean ± standard error of mean (SEM). For the postprandial plasma responses incremental areas under the curve (iAUC’s) were calculated using the trapezoid method for the CD and both HD-groups [27]. The effect of diet, period and the interaction between factors for the fasted blood plasma metabolites and the iAUC’s across all treatments were analyzed using analysis of variance (ANOVA, PROC GLM procedure of SAS). Post-hoc Tukey tests for multiple comparisons were performed to do pair-wise comparisons. All *p*-values were considered significant if *p* < 0.05. Statistical analyses were performed with SAS 9.3 (SAS 9.3; 2002–2010 by SAS Institute Inc., Cary, NC, USA).

## 3. Results

### 3.1. Fasting Plasma Concentrations of Biomarkers for Lipid and Glucose Metabolism

Baseline subject characteristics, previously published [11], showed no difference between groups. Fasted blood was collected from all subjects before each MC (Table 2). Fasting glucose significantly increased over the intervention periods (*p* = 0.04), particularly in the HD-group but comparing all three groups showed no effect of diet (*p* = 0.90). Fasting insulin showed neither a period (*p* = 0.25), nor a diet effect (*p* = 0.81). Fasting TG concentration was significantly different between diets (*p* < 0.01), with no effect of the period (*p* = 0.89). Both HD diets showed a significantly lower fasting TG concentration compared with the CD-diet (HP: *p* < 0.01; NP: *p* = 0.04), but no difference was observed between the two HD diets. Fasting FFA decreased over the intervention period in all groups (*p* < 0.01) but comparing the three groups showed no effect of diet (*p* = 0.65). Fasting β-hydroxybutyrate decreased over the intervention period (*p* = 0.02), particularly in the HD-groups, but no effect of diet was observed (*p* = 0.45). Fasting glycerol, HDL, and total cholesterol showed neither a period nor a diet effect.

### 3.2. Postprandial Blood Biomarkers in Response to a Meal Challenge (MC)

The 360 min iAUCs of glucose showed a trend for a difference between the three diets (*p* = 0.05), but no effect of the period. Within the HD-adapted group, the glucose response was significantly higher after adaptation to the HP diet compared to the NP diet (iAUC 122.7 ± 13.4 mmol/L vs. 86.6 ± 14.2 mmol/L respectively; *p* = 0.03) (Figure 2). For insulin, neither an effect of diet nor of period was detected (diet: *p* = 0.48; period: *p* = 0.17; data not shown). Also, no difference within the HD diet between the NP and the HP content was found (iAUC 3364.4 ± 552.2 vs. 3040.5 ± 521.7 mmol/L respectively; *p* = 0.37). 

The 360 min iAUCs of plasma TG revealed a significant increase with the intervention period (*p* < 0.01) but no effect of diet (*p* = 0.24) and no difference within the HD-adapted group between the NP and the HP condition (iAUC 125.5 ± 18.9 mmol/L vs. 111.1 ± 14.1 mmol/L respectively; *p* = 0.17) (Figure 2). The suppression of plasma FFA in response to the MC was significantly different between the three diets (*p* = 0.04) with no effect of the period (*p* = 0.63). iAUC was significantly smaller after the HP diet compared to the CD diet (*p* = 0.01). Within the HD-adapted group FFA suppression tended to be smaller in the HP compared to the NP condition (360min iAUC −47.0 ± 7.5 mmol/L vs. −58.7 ± 9.5 mmol/L respectively; *p* = 0.06) (Figure 2).

The iAUCs of plasma glycerol was significantly different between the three diets (*p* = 0.003), while no effect of period was observed (*p* = 0.46). It was significantly lower with the HP and NP diet compared to the CD (*p* < 0.01 *p* < 0.01 respectively). Within the HD-adapted group, it was not different between the HP and the NP condition (iAUC 1435.9 ± 324.2 µmol/Lvs. 1733.2 ± 427.6 µmol/L respectively; *p* = 0.55) (Figure 2). The iAUCs of plasma total cholesterol, HDL cholesterol and β-hydroxybutyrate responses to the MC showed no period or diet effect and no difference within the HD-adapted group between the HP and the NP diet (data not shown).

### 3.3. Urinary and Plasma Metabolites Markers of High Caloric Diet with High or Low Protein Content

Fasting plasma and urine samples were analyzed using the RP and HILIC techniques and IC-DA was applied to the MS data. For plasma samples, no clear discrimination between groups was observed, neither for diet nor for period (data not shown). In contrast, for 24 h urine the HD groups were separated from the CD-group along IC1, IC2 and IC5, (as shown in the proportions plot Figure 3A), after source signal extraction by IC-DA based on 3 groups (QC, CD and HD), from the data obtained by RP chromatography. An IC-DA based on the groups associated with the types of intervention (QC, HP-NP, NP-HP, CD) did not discriminate clearly between the two sub-groups NP-HP (HD-1) and HP-NP (HD-2) that constitute the HD group (Figure 3B) whereas when the IC-DA was done based on the 3 groups (QC, HP and (CD + NP), the HP group was separated from the (CD + NP) group (Figure 3C).

The predictive performance of the IC-DA models was tested by group permutations and comparing the F-distributions of the corresponding Wilks’ lambdas, and the F-values obtained for the true groups was higher than all the F-values calculated for the permuted groups for the first and third models, with empirical *p* values of 0, thus confirming the validity of these models The IC-DA performed on the data obtained by the HILIC technique also discriminated the HD group from the CD group (Figure 4A). Again, the IC-DA based on intervention (QC, HP-NP, NP-HP, CD) (Figure 4B) did not separate HP-NP from NP-HP, while the IC-DA based on the 3 groups (QC, HP and (CD + NP) allow the separation of the HP group from the CD and NP groups (Figure 4C).

The S-Plot [28] was used to select the most relevant variables for the separation of groups. The selection of discriminant variables was based on correlations and covariances, and contributions to the ICs which are greater than +/− 3 standard deviations of all the contributions to each component. IC-DA revealed that 13 features (detected by RP and/or HILIC) were associated with the diets consumed. The detailed information on each feature is reported in Table 1.

Two compounds, annotated as octenoylcarnitine and decatrienoylcarnitine were specifically associated with the normal protein hypercaloric high-fat diet. The relative intensities and therefor urinary excretion of these compounds are significantly higher after adaptation to the NP diet compared to the HP and CD diet (Figure 5).

Eleven compounds were specifically associated with the HP hypercaloric high-fat diet. Among these compounds’ carnitine, 3 short-chained acylcarnitines (C2, C3, C4 and C5 acylcarnitines) and uric acid were identified (Table 1; Appendix A). Three other compounds, namely oxoguanine, C5:1-M and C10:2-acylcarnitines were tentatively annotated. The *m*/*z* 328.2485 was tentatively annotated as an unknown acylcarnitine, and m/z 181.0711 showed the MS-MS fragmentation pattern identical to theophylline (although the retention time was different) suggesting the presence of a compound with a structure similar to theophylline.

The excretion of carnitine and short-chained acylcarnitines was significant higher (*p* < 0.05) in urine of participants after consuming the HP hypercaloric diet, compared to the control diet or the normal protein hypercaloric high-fat diet (Figure 6A). The excretion of heterocyclic nitrogen-containing metabolites (oxoguanine, uric acid and theophylline-like metabolite) was significantly lower (*p* < 0.05) after adaptation to the HP condition as compared to CD and NP condition (Figure 6B). No significant effect of period was observed for all these features except for 2-methylbutyroylcarnitine (*p* = 0.009), which increased significantly after the HP diet in P2, but not in P1. In addition to the aforementioned compounds, 19 others were identified/annotated in the HP group, but none showed significant differences in excretion after HP diet consumption (Appendix A).

Interestingly, the ^13^C enrichment of urine was higher in the HP condition as compared to control group and the NP condition (*p* = 0.005), whatever the period, whereas in the NP condition the enrichment was similar to the control group (Figure 7 and Appendix A).

## 4. Discussion

This study addressed the consequences of 2 weeks adaptation to a hypercaloric high-fat diet with either a high- or normal-protein content, compared to a parallel weight-maintaining control diet, on fasting and postprandial plasma and 24 h urine markers of glucose and lipid metabolism. Increasing protein within a hypercaloric diet was previously shown to improve fasting lipid metabolism and prevent the increase in intra-hepatic lipid content and body fat mass [10,11]. The current study indicates that a HP-HD diet, compared to a NP-HD diet, induced a slightly weaker suppression of circulating FFA after a mixed meal, while the postprandial glucose response was higher. The results on urinary metabolomic profiles showed a higher excretion of carnitine, short-chain acylcarnitines and decadienoylcarnitine and a lower excretion of medium chain acylcarnitines (decatrienoylcarnitine and octenoylcarnitine) after adaptation to the HP-HD diet compared to a NP-HD diet. Finally, the enrichment of ^13^C in urine was higher after the HP-HD diet.

In the present study, adaptation to the HP diet compared to the NP diet partly affected glucose and lipid metabolism in postprandial but not fasting state. Participants who had previously consumed the HP diet showed a slightly higher iAUC for glucose with no change on insulin response after the mixed meal challenge. Additionally, the suppression of circulating FFA tended to be smaller after the HP when compared with the NP condition after a meal of the same macronutrient composition. The postprandial glycerol response showed a smaller decrease after the HD compared to the CD with no significant difference between the NP-HD and the HP-HD diet. Other markers of lipid metabolism including HDL cholesterol, total cholesterol and β-hydroxybutyrate were not altered after adaptation to the different diets. These results agree with observations that high dietary protein acutely affects lipid metabolism, with lowering effects on cholesterol synthesis and lipogenic enzymes [10,29] and a potential reduction of lipidemia [30,31,32]. High protein diets can stimulate postprandial gluconeogenesis in the liver, with a reduced inhibition of glucose production by the meal [33,34]. They also have the potential to acutely stimulate postprandial insulin secretion [35]. This in turn can stimulate the uptake of fatty acids by tissues, in great part by adipose tissue and muscle that highly express lipoprotein lipase (LPL), and also block lipolysis by inhibiting hormone-sensitive lipase [36]. A slight decrease in observed insulin sensitivity [5] may mediate the combined effects on lipid and glucose metabolism in the high-protein condition, but might be a transient situation. Moreover, a HP diet enhances also glucagon secretion leading to the stimulation of gluconeogenesis and lipolysis [10]. Rather than reflecting a lower insulin sensitivity induced by HP intake, our results may reflect a lower hepatic lipogenesis and a lower FFA oxidation due to a higher protein oxidation [10,37]. Indeed, when protein intake is increased and leads to a level of protein-derived dietary AA above the maximum level possibly channeled to protein synthetic pathway, the resulting excess AAs are deaminated and the AA-derived carbon skeletons are, according to the AAs, converted to acetyl-CoA or tricarboxylic acid cycle (TCA) intermediates (succinyl-CoA, fumarate, α-keto glutarate, and oxaloacetate) and channelled into energy pathways, either the TCA cycle or ketogenesis. The increased influx of AA into the TCA cycle leads to cellular accumulation of acetyl-CoA, which allosterically inhibits pyruvate dehydrogenase and the conversion of pyruvate produced by glucose catabolism to acetyl CoA, that promotes mitochondrial export of carbon metabolites acting as negative regulators of fatty acid oxidation.

A non-targeted urinary metabolomics approach led to the detection of a panel of potential biomarkers for protein intake and the metabolic state induced by the different diets, including short- and medium-chain acylcarnitines. Carnitine is provided from the diet and also endogenously synthesized from lysine and methionine. Higher excretion of creatine and carnitine was previously described after high-protein intake [38]. In the present study, dietary protein was of mixed origin (dairy, animal, and plant sources), with the contribution of dairy protein more prominent in the HP condition (55% compared with 30% for both other diets) and, therefore, increased carnitine could be related to higher animal protein intake.

When comparing the HP-HD to the NP-HD and CD condition, a higher urinary excretion of several short-chain acylcarnitines and decadienoylcarnitine was observed. Carnitine and high concentrations of short-chain acylcarnitines (acetylcarnitine, butyrylcarnitine, and methylbutyrylcarnitine) investigated by metabolomic profiling in plasma and liver biopsies were previously identified as risk biomarkers in patients with NAFLD or non-alcoholic steatohepatitis [39,40]. Our results also showed a higher excretion of two medium chain acylcarnitines (decatrienoylcarnitine and octenoylcarnitine) after adaptation the NP-HD compared to the HP-HD and the eucaloric control diet. As previously described, the NP-HD tended to induce higher intrahepatic lipids (IHL) compared to the HP-HD [11]. Therefore, we can speculate that there is an association between the higher IHL and the higher excretion of medium chain acylcarnitines in the present analysis. The higher amount of urinary short-chain acylcarnitines after the HP-HD could in turn be associated with the prevention of this NP-HD induced increase in IHL, which was previously described [10,11]. Thus our data suggest that rather than short-chain acylcarnitine, high concentrations of medium chain acylcarnitines could be a signature of the state of higher IHL. Moreover, branched-chain amino acids which are increased in patients with non-alcoholic steatohepatitis, were not detected even with HILIC, suggesting that high concentrations of branched-chain plasma amino acids are more related to mitochondrial dysfunction than to protein intake [41].

Carnitine has a key role in mitochondrial oxidation of long-chain fatty acids through the action of specialized acyltransferases and thus may be involved in the removal of excess acyl groups and peroxisomal fatty acid oxidation [37,42]. Carnitine acetyltransferase (CrAT) generates short- to medium-chain carnitines, thereby acting as critical modulators of matrix acyl-CoA concentrations and control of the fate of both FA and pyruvate oxidation, thereby potentially contributing to tissue metabolic fuel “choice” [42]. The role of protein intake and thereby amino acid fuel supply in the fate of both FA and pyruvate oxidation is largely unknown. Moreover, carnitine has multiple roles in mammalian metabolism, including the shuttling of beta-oxidation chain-shortened products out of peroxisomes in the liver and the modulation of the acyl-CoA/CoASH ratio [43]. In this study, excretion of carnitine and short chain acylcarnitine (C2, C3 and C4) increased with the increase of protein intake. This suggests that FA oxidation increased in the liver but was incomplete and through the stimulation of carnitine synthesis C2, C3 and C4 beta-oxidation chain-shortened products were shuttled out of the liver and excreted in urine. This mechanism could be partly responsible for the prevention of liver lipid accumulation. It has been proposed that increased concentration of short chain acylcarnitine species could be related to muscle branched chain amino acids catabolism but not fatty acid catabolism [44]. However, to the best of our knowledge, no study has investigated branched-chain amino acids metabolic fluxes proving that short-chain acylcarnitine arises from muscle branched-chain amino acids catabolism. Interestingly, we have also shown a higher ^13^C-enrichment in 24 h urine after HP-HD intake which may be related to the higher concentrations of short-chain acylcarnitine species. Also, rather than reflecting the dietary intake (lipids are naturally depleted in ^13^C) [45] it could indicate the fixation of bicarbonate (naturally ^13^C-enriched) by carbamoyl-phosphate (CP) synthesis to feed the urea cycle and the anaplerotic pathway [46].

Interestingly, medium-chain (decatrienoylcarnitine and octenoylcarnitine) acylcarnitine urinary excretion were higher following NP-HD intake compared to the HP-HD and CD suggesting here, again, an incomplete oxidation of FA. An increase of medium-chain acylcarnitines has been proposed to reflect inefficient or incomplete long-chain fatty acids β-oxidation. Unlike mitochondria, peroxisomal oxidation is incomplete and cannot chain-shorten fatty acids beyond six carbon residues. This leaves a medium-chain acyl-CoA derivative as well as acetyl-CoA that can be released into the cytosol as unbound carbon moieties or as carnitine derivatives. One can hypothesize that higher dietary protein induces higher mitochondrial rather than peroxisomal oxidation. In a rodent model, an elevated mitochondrial biosynthesis in response to a HP diet compared to a NP diet was reported [37]. Interestingly, Noland et al. 2007 [47] reported that mitochondrial oxidation of various lipid substrates (palmitate, octanoate and octanoylcarnitine) diminished by 50% in rodent model of obesity-associated insulin resistance which may contribute to the excessive lipid accumulation seen in this tissue. Altogether, the present results suggest that HP intake may improve complete medium chain fatty acids oxidation. It may thereby prevent hepatic fatty acid accumulation induced by HD intake and energy accumulation through the increase of carnitine synthesis and thereby urinary excretion of short chain acylcarnitines. The fate of short- and medium-chain fatty acids could explain the non-stimulation of ketogenesis and higher postprandial circulating FFA following HP intake.

Including only young healthy men and women in the trial gave an opportunity to study the metabolism without interference of metabolic disorders like impaired insulin sensitivity, which affects both glucose and lipid metabolism. Such a population is highly metabolic flexible and has the ability to adapt fuel oxidation to fuel availability [48]. The results also showed that urinary metabolomic is a sensitive approach to identify short- chain and medium-chain acylcarnitines as biomarkers of the metabolic state induced by an adaptation to a HP-HD and a NP-HD diet, possibly including hepatic lipid storage. The higher excretion of acyls through their conjugation to carnitine could facilitate the elimination of excess dietary fat through short-chain urinary acylcartine and restore complete medium-chain fatty acid oxidation as suggested by the lower urinary excretion of short-chain acylcarnitine. These medium-chain acylcarnitines could, therefore, be early biomarkers of hepatic lipid accumulation.

## 5. Conclusions

Our study suggests a role of urinary acylcarnitine as biomarkers of protein intake and hepatic lipid accumulation. Urinary carnitine and short-chain urinary acyl-carnitines are higher in response to a HP-HD diet and decatrienoylcarnitine and octenoylcarnitine medium-chains acyl-carnitine are higher after the NP hypercaloric high-fat diet. ^13^C excretion was higher after a HP-HD diet. Further studies are needed to understand the metabolic fate of high caloric intake depending on the protein content of the diet in order to clarify the role of conjugation to carnitine in the control of energy homeostasis and steatosis.

## Figures and Tables

**Figure 1 nutrients-13-01191-f001:**
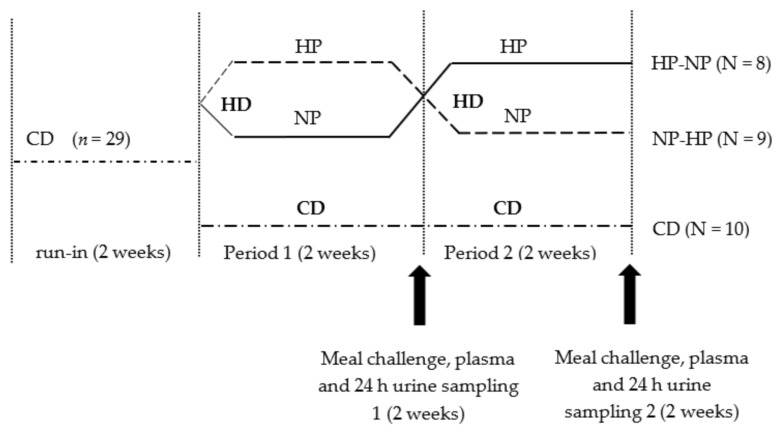
Study design. All participants started on a two-week run-in period on a weight maintaining control diet (CD, 27.8 En% fat; 16.9 En% protein; 55.3 En% carbohydrate). Thereafter, participants were randomly assigned to either the high-fat-hypercaloric diet group (HD; *n* = 19) or to the reference group consuming a control diet (CD; *n* = 10) for the following 4 weeks of intervention. Participants in the HD-group were overfed with 2 MJ per day. The HD intervention consisted of a of a 2 × 2-week cross-over design with two separate periods: a 2-week high-protein intervention (HP, 37.7 En% fat; 25.7 En% protein; 36.6 En% carbohydrates) and a 2-week normal-protein intervention (NP; 39.4 En% fat; 15.4 En% protein; 45.2 En% carbohydrates). We collected 24 h urine samples at the end of periods 1 and 2, a meal challenge with blood sampling was performed after period 1 and 2. HD—high-fat hypercaloric diet; CD—control diet; HP—high protein condition (within high-fat hypercaloric diet); NP, normal protein condition (within high-fat hypercaloric diet).

**Figure 2 nutrients-13-01191-f002:**
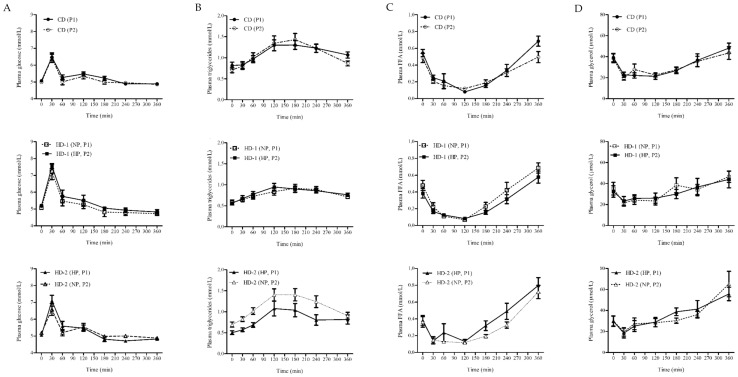
Response AUC of plasma during the meal challenge of: (**A**) glucose; (**B**) triglycerides (TG); (**C**) free fatty acids (FFA) and(**D**) glycerol, in all dietary groups; participants where adapted to either a control diet (CD) for 4 weeks (*n* = 10) or a high fat hypercaloric diet (HD) differing in protein content, normal (NP) vs. high (HP) in random order (NP-HP), *n* = 8; HP-NP, *n* = 9), P1 and P2. Data are mean ± standard error (SE).

**Figure 3 nutrients-13-01191-f003:**
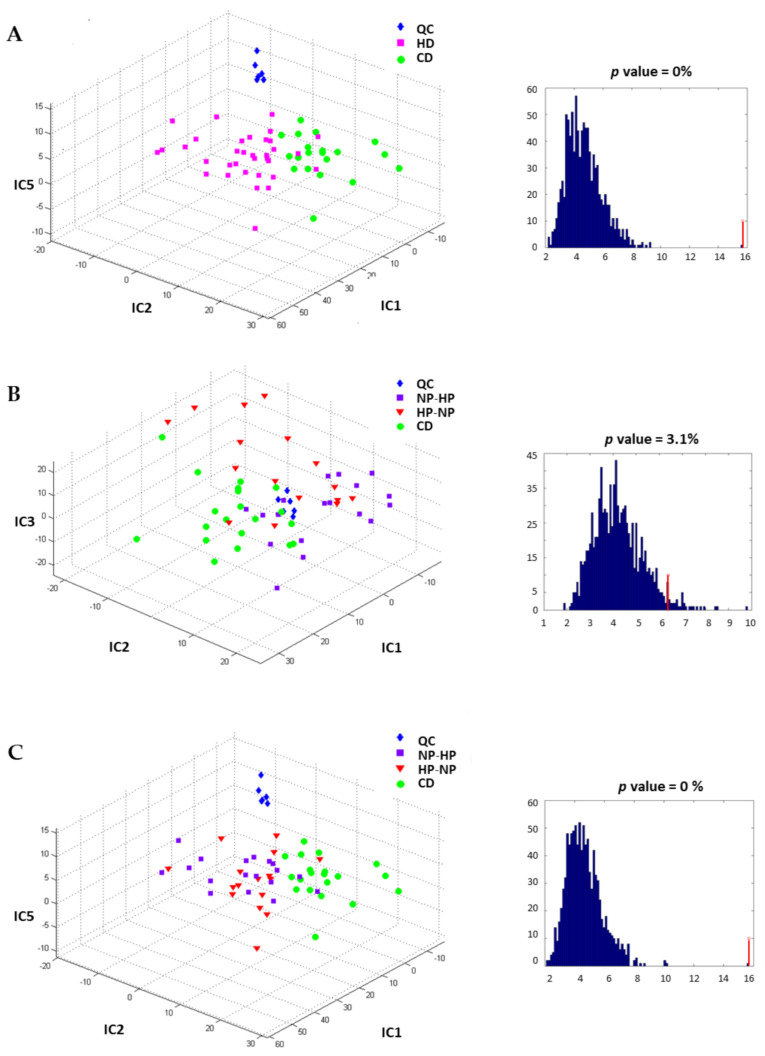
Independent components discriminant analysis proportions plot calculated after source signal extraction for the data obtained by RP chromatography and corresponding F-value associated with the Wilks’ lambda characterizing the discrimination of the predefined groups: (**A**) to discriminate the fat content of the diet. Separation of the HD groups from the CD-group based on IC1, IC2 and IC5; (**B**) to discriminate the protein content diet sequences, partial separation of the NP-HP group with HP-NP group based on IC1, IC2 and IC3; (**C**) to discriminate the diets, whatever the intervention period. Separation of the HP, NP and CD groups based on IC1, IC2 and IC5. CD: control diet; HD: high-fat, hypercaloric diet whatever the protein content of the diet; NP-HP: NPHD at P1 and HPHD P2 period; HP-NP: HPHD at P1 and NPHD P2 period; HP: HPHD at either P1 or P2 period; NP: NPHD at either P1 or P2 period; QC: quality controls.

**Figure 4 nutrients-13-01191-f004:**
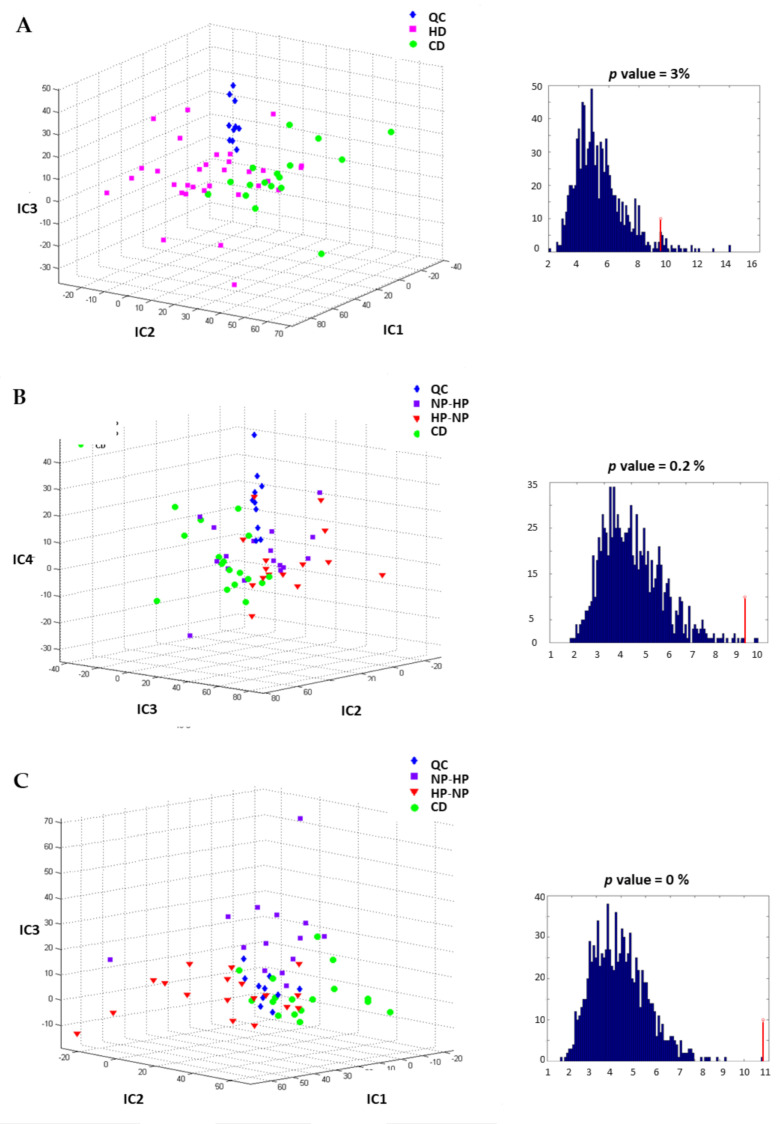
Independent components discriminant analysis proportions plot calculated after source signal extraction for the data obtained by HILIC chromatography and corresponding F-value associated with the Wilks’ lambda characterizing the discrimination of the predefined groups: (**A**) to discriminate the fat content. Separation of the HD groups from the CD-group based on IC1, IC2 and IC2; (**B**) to discriminate the protein content diet sequences, poor separation of the NP-HP group with HP-NP group based on IC2, IC3 and IC4; (**C**) to discriminate the diets, whatever the intervention period. Partial separation of the HP, NP and CD-groups based on IC1, IC2 and IC3. CD: control diet; HD: high-fat, hypercaloric diet whatever the protein content of the diet; NP-HP: NPHD at P1 and HPHD P2 period; HP-NP: HPHD at P1 and NPHD P2 period; HP: HPHD at either P1 or P2 period; NP: NPHD at either P1 or P2 period; QC: quality controls.

**Figure 5 nutrients-13-01191-f005:**
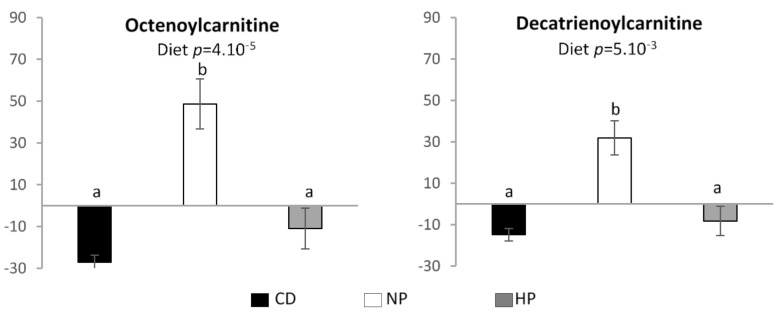
Relative signal intensity of features in urine associated with the NP high-fat hypercaloric diet. Data are mean ± SEM. The *p* value is indicated on each graph, when the difference is significant, for the diet and period effect (analysis of variance (ANOVA)). When the diet effect is significant, different letters indicate significant differences between groups. CD—control diet; NP—normal protein, high-fat, hypercaloric diet; HP—high protein, high-fat, hypercaloric diet.

**Figure 6 nutrients-13-01191-f006:**
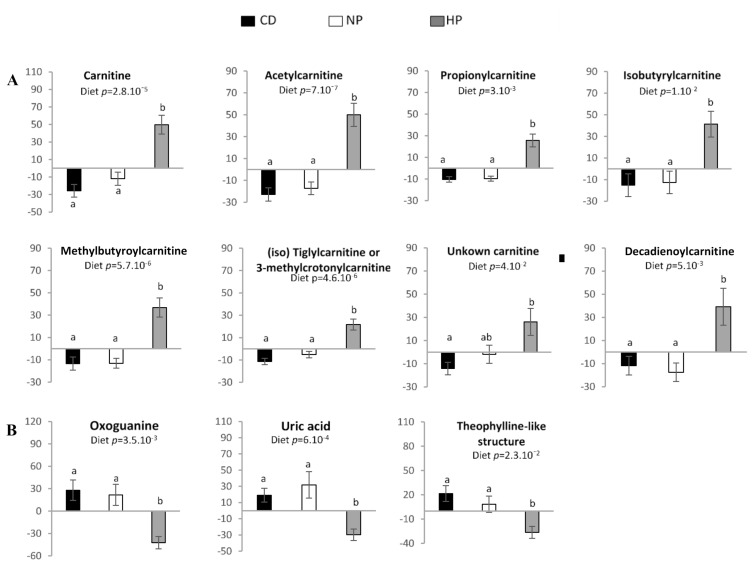
Relative signal intensity of features in urine associated with the HP high-fat hypercaloric diet: (**A**) Carnitine and short-chained acylcarnitines; (**B**) heterocyclic nitrogen-containing metabolites. Data are mean ± SEM. The *p* value is indicated on each graph, when the difference is significant, for the diet and period effect (ANOVA). When the diet effect is significant, different letters indicated significant differences between groups. CD—control diet; NP—normal protein, high-fat, hypercaloric diet; HP—high-protein, high-fat, hypercaloric diet.

**Figure 7 nutrients-13-01191-f007:**
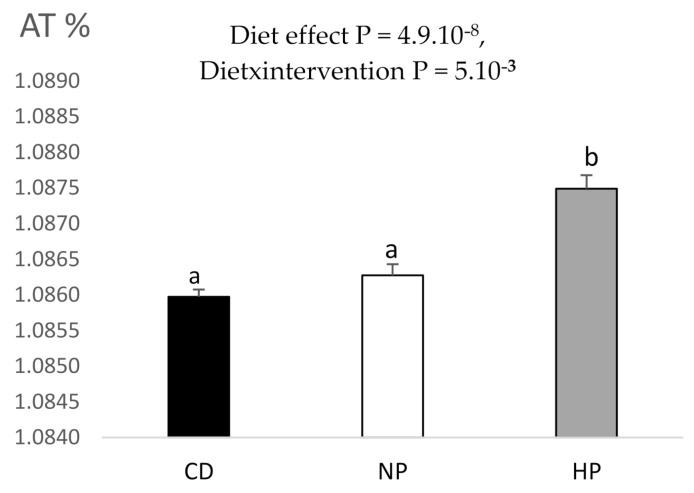
^13^C enrichment observed in 24 h urines of participants. Data are mean ± SEM. The *p* value is indicated on the graph, when the difference is significant, for the diet and period effect (ANOVA). When the effect is significant, different letters indicated significant differences between groups. CD, control diet; NP, normal protein, high-fat, hypercaloric diet diet; HP, high-protein, high-fat, hypercaloric diet.

**Table 1 nutrients-13-01191-t001:** Mass spectrometry (MS) characteristics of diet-associated discriminant features detected by reverse-phase (RP) chromatography and hydrophilic interaction chromatography (HILIC).

R_T_, min (RP/HILIC)	Experimental *m*/*z*	RP/HILIC	Elemental Formula	MS/MS Fragment Ions	Collision Energy, eV	MSI MI Level	Compound
0.48/5.40	162.1114	RP/HILIC	C_7_H_16_NO_3_	103.0396, 85.0295	20	1	carnitine
0.50/4.49	204.1216	RP/HILIC	C_9_H_18_NO_4_	144.1015, 85.0291	15	1	acetylcarnitine (C2)
0.59/3.53	218.1374	RP	C_10_H_20_NO_4_	159.0656, 144.1039, 85.0289	15	1	propionylcarnitine (C4)
0.66	169.0355	RP	C_5_H_5_N_4_O_3_	152.0097, 141.0423, 126.0302, 98.0357, 96.0194, 70.0409, 69.0080	20	1	uric acid
0.70	167.0559	RP	C_6_H_7_N_4_O_2_	149.0470, 124.0518, 110.0356, 96.0554, 82.0407	20	2	oxoguanine
0.70	181.0711	RP	C_7_H_9_N_4_O	163.0618, 135.0715, 124.0516, 67.0287	20	3	theophylline-like structure
0.70	244.1525	RP	C_12_H_22_NO_4_	185.0824, 144.1022, 85.0288	20	2	(iso)tiglylcarnitine or 3-methylcrotonylcarnitine (C5:1)
0.70/2.90	232.1529	RP/HILIC	C_11_H_22_NO_4_	173.0833, 144.1036, 128.0701, 85.0285	15	1	isobutyrylcarnitine (C4)
0.70	246.1683	RP	C_12_H_24_NO_4_	187.0973, 144.1012, 85.0289	15	2	2-methylbutyroylcarnitine (C5)
3.97/1.72	286.2013	RP/HILIC	C_15_H_28_NO_4_	227.1272, 144.1019, 125.0972, 97.1009, 85.0290	20	2	octenoylcarnitine (C8:1)
4.35/1.48	310.2013	RP/HILIC	C_17_H_28_NO_4_	251.1277, 149.0961, 144.1022, 121.1018, 93.0714, 85.0294	20	2	decatrienoylcarnitine (C10:3)
5.13	312.2169	RP	C_17_H_30_NO_4_	253.1436, 151.1120, 144.1032, 123.1174, 85.0290	20	2	decadienoylcarnitine (C10:2)
5.68	328.2485	RP	C_18_H_34_NO_4_	269.1742, 144.1016, 85.0284	20	--	unknown acylcarnitine

eV: electron Volts, HILIC: hydrophilic interaction chromatography, MS: Mass spectrometry, MSI MI: level—Metabolomic standards initiative metabolite identification level, RP: reverse-phase chromatography Rt: retention time.

**Table 2 nutrients-13-01191-t002:** Fasted plasma concentrations, determined from venous blood in all groups.

	CD		NP-HP		HP-NP		*p*-Value	
	P1	P2	P1 (NP)	P2 (HP)	P1 (HP)	P2 (NP)	Diet	Period
Glucose (mmol/L)	5.07 ± 0.04	5.08 ± 0.06	4.98 ± 0.11	5.14 ± 0.12	4.97 ± 0.13	5.09 ± 0.11	0.90	0.04
Insulin (mU/L)	4.0 1± 0.62	4.22 ± 0.85	4.74 ± 1.14	4.93 ± 1.13	3.08 ± 0.57	3.74 ± 0.62	0.81	0.25
TG (mmol/L)	1.03 ± 0.10	0.85 ± 0.08	0.69 ± 0.06	0.70 ± 0.04	0.61 ± 0.06	0.84 ± 0.06	0.003 *	0.89
FFA (mmol/L)	0.40 ± 0.07	0.25 ± 0.03	0.41 ± 0.08	0.22 ± 0.05	0.32 ± 0.08	0.21 ± 0.03	0.65	0.003
β-HB (mmol/L)	0.07 ± 0.02	0.05 ± 0.01	0.09 ± 0.03	0.05 ± 0.01	0.12 ± 0.04	0.04 ± 0.01	0.45	0.02
Glycerol (umol/L)	40.8 ± 4.43	34.9 ± 4.69	45.2 ± 6.84	34.2 ± 6.22	36.0 ± 6.37	37.2 ± 6.57	0.45	0.18
Total Cholesterol (mmol/L)	4.79 ± 0.30	4.33 ± 0.25	4.69 ± 0.42	4.35 ± 0.18	4.98 ± 0.32	5.51 ± 0.59	0.20	0.61
HDL Cholesterol (mmol/L)	1.19 ± 0.09	1.11 ± 0.07	1.33 ± 0.08	1.32 ± 0.15	1.24 ± 0.09	1.25 ± 0.07	0.20	0.52

Data are mean ± SEM, CD: *n* = 10 NP-HP: *n* = 8, HP-NP: *n* = 9. * Significant difference HP vs. CD: *p* = 0.0009; NP vs. CD *p* = 0.04. CD—control diet; HP—high-protein condition (within high-fat hypercaloric diet); NP—normal-protein condition (within high-fat hypercaloric diet); P1—period 1; P2—period 2.

## Data Availability

Data described in the manuscript, code book, and analytic code will be made available upon request pending application and approval.

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
