# Peer review of "Urinary Medium-Chained Acyl-Carnitines Sign High Caloric Intake whereas Short-Chained Acyl-Carnitines Sign High -Protein Diet within a High-Fat, Hypercaloric Diet in a Randomized Crossover Design Dietary Trial"

_nutrients, 2021, doi:10.3390/nu13041191_

Round 1

Reviewer 1 Report

This study aimed to identify biomarkers of glucose and lipid metabolism in fasting and postprandial plasma and 24-h urine of humans consuming a hypercaloric diet with either high-protein or normal protein content and those consuming a weight maintaining control diet. The experiments were carefully designed and the manuscript was well-written. I have a few minor suggestions/questions:

  1. In the section “postprandial blood biomarkers in response to a MC”, it is stated that “a significant correlation was observed between iAUC of glucose and insulin in response to the MC”. However, the data for the iAUC of insulin response was not shown. Can you explain how you calculate the correlation between iAUC of glucose and insulin response?
  2. In Figure 2, within each sub-column, there were ABCs showing different graphs. However, in the Figure legend, A, B, C, D was assigned to different objects (A for glucose, B for triglyceride, C for free fatty acids and D for glycerol). It is very confusing to the readers and please modify the labels to make it clear.
  3. In the section “Urinary and plasma metabolites markers of high caloric diet with high or low protein content”, what do IC1, IC2, IC3 and IC5 mean? I noticed in Fig. 3A and 3C, IC1, IC2 and IC5 were used to indicate each axis whereas in Fig. 3B, IC1, IC2 and IC3 were used.
  4. For graphs on the right panel of Fig.3, the label on top of each graph was not readable, can you make the font size larger? Additionally, a red line was shown in the graph, what does it stand for?
  5. Please spell out the full name for NASH and BCAA.
  6. The page number and line numbers were completely messed up. Please fix them.

Reviewer 2 Report

Here the authors presented an elegant study about the excretion of medium-chained acyl carnitines and short-chained acyl-carnitines after high or normal protein diet administration for two weeks. They showed that the higher excretion of short-chained acyl-carnitines could facilitate the elimination of excess fat of the high protein diet, reducing fat accumulation. On the other hand, they showed that the higher excretion medium-chained acyl-carnitines could be considered an early biomarker of hepatic lipid accumulation.

In my opinion, the paper is generally well written and it proceeds logically. The material and methods are explained in detail and the results are clearly presented. The discussion is very well argued and the conclusions are in line with the results. I suggest only a minor revision to eliminate some oversights in the text.
